# Spasm of Near Reflex in a Patient with Autism Spectrum Disorder: A Case Report

**DOI:** 10.3390/reports6030038

**Published:** 2023-08-08

**Authors:** Satoshi Ueki, Yukari Hasegawa, Tetsuhisa Hatase, Takako Hanyu, Jun Egawa, Atsushi Miki, Takeo Fukuchi

**Affiliations:** 1Division of Ophthalmology and Visual Science, Graduate School of Medical and Dental Sciences, Niigata University, Asahimachi-dori 1-757, Chuo-ku, Niigata 951-8510, Japantfuku@med.niigata-u.ac.jp (T.F.); 2Imai Eye Clinic, Honcho 3-2-1, Niigata 957-0054, Japan; 3Hanyu Clinic, Igarashi-higashi 1-1-15, Nishi-ku, Niigata 950-2045, Japan; 4Department of Psychiatry, Graduate School of Medical and Dental Sciences, Niigata University, Asahimachi-dori 1-757, Chuo-ku, Niigata 951-8510, Japan; 5Department of Ophthalmology, Kawasaki Medical School, Matsushima 757, Okayama 701-0192, Japan

**Keywords:** spasm of near reflex, autism spectrum disorder, microfluctuations in accommodation

## Abstract

Spasm of near reflex (SNR) involves intermittent spasm of one or more of the three near reflex components. Psychiatric disorders are one cause of SNR. We describe a patient with SNR diagnosed with autism spectrum disorder (ASD). A 36-year-old male with esotropia since childhood was referred due to headache and dizziness. The alternate prism cover test showed 30 prism diopters at both near and distant fixation. Four months after his first visit, he was diagnosed with ASD. Twenty-nine months after his first visit, he underwent strabismus surgery to treat concomitant esotropia. Postoperatively, the angle of strabismus improved but remained variable. Because the angle of strabismus varied, we suspected SNR; the diagnosis was performed after evaluating the patient’s microfluctuations in accommodation with Speedy-K. However, it was difficult to distinguish convergence spasm from concomitant esotropia in this patient because he has had a history of esotropia since childhood. In a patient with concomitant esotropia, if the symptoms are not exclusively due to strabismus, SNR should be suspected. Although the relationship between SNR and the pathology of ASD is unknown, it is possible that patients with ASD are more likely to develop SNR.

## 1. Introduction

Spasm of near reflex (SNR) is a condition that presents with spasm of one or more of the three components of the near reflex: convergence, accommodation, and pupil constriction. There are usually intermittent subjective symptoms and variable clinical findings [1]. Therefore, an accurate diagnosis of SNR is hard to make. There are four main causes of SNR: psychiatric disorders (e.g., stress, anxiety disorders, personality disorders), head trauma, organic diseases, and other [2]. Organic diseases include multiple sclerosis, Chiari malformation, neurofibroma, pituitary adenoma, labyrinthitis, elevated intracranial pressure, vestibular neuritis, metabolic encephalopathy, stroke, epilepsy, dorsal midbrain syndrome, and cerebral aneurysm [2]. Other include post laser in situ keratomileusis, intermittent exotropia, excessive or prolonged near work, instrument myopia, hereditary cause, post myelography, and dissociation of binocularity [2]. The incidence of SNR in patients who visited a single binocular vision and orthoptics clinic was 5.1% [3]. We encountered an adult patient with concomitant esotropia, SNR, and autism spectrum disorder (ASD). Because he has had concomitant esotropia since childhood, it was difficult to distinguish convergence spasm from concomitant esotropia. We report the important clinical symptoms of SNR to suspect the disease based on the findings in our patient and previous reports. Although there are no reports describing a patient with both SNR and ASD and the incidence of SNR in patients with neurodevelopmental disorders such as ASD is undetermined, reports have described a relationship between near reflex and ASD [4,5]. We discuss the relationship between SNR and ASD. The objectives of this case report are to propose tips to accurately diagnose SNR and to present an in-depth discussion on whether patients with ASD might develop SNR.

## 2. Detailed Case Description

A 36-year-old male patient was referred to the Department of Ophthalmology at the Niigata University Medical and Dental Hospital with headache and dizziness. He thought that strabismus was the cause of his symptoms. He had a history of esotropia that began during childhood. Details of this condition were unknown. His corrected visual acuity was 20/20 with +1.25 diopter (D) in the right eye and 20/20 with +1.25 D in the left eye. Cycloplegic refraction was not measured. The cover–uncover test revealed esotropia at both near and distance. The alternate prism cover test revealed the angle of strabismus was approximately 30 prism diopters (PD) at both near and distance. There were no limitations in abduction during either duction or version. Synoptophore (HAAG-STREIT UK, Bishop’s Stortford, UK) revealed the absence of simultaneous perception. Because the patient felt an amelioration of his symptoms when wearing glasses with Fresnel membrane prisms (15 PD base out for each eye), he was instructed to start wearing the glasses. We thought that the glasses could reduce the degree of change in alternating visual images. Twelve months after his first visit, he returned to our department. He was not wearing glasses with Fresnel membrane prisms. Thus, he was instructed to wear glasses with built-in prisms (6 PD base out for each eye). Fifteen months after the initial visit, the patient complained that wearing the glasses with built-in prisms reduced his symptoms. Sixteen months after his initial visit to our department, he was referred to the Department of Psychiatry at the Niigata University Medical and Dental Hospital due to decreased thinking ability because of pain in his left ear. After a close examination, he was diagnosed with ASD and somatic symptom disorder (he complained of pressure on his face). The patient was diagnosed in accordance with the Diagnostic and Statistical Manual of Mental Disorders, Fifth Edition, (DSM-5) criteria for ASD by an experienced child psychiatrist. DSM-5 criteria for ASD include persistent deficits in social interaction across multiple contexts, social communication impairments, and restricted, repetitive patterns of behavior. Specifically, the patient had “difficulty with interpersonal interactions since early childhood and was unable to make friends even as a student. Even during medical examinations, he had difficulty making eye contact with others, which prevented him from deepening his emotional interactions”. Episodes indicative of social communication impairments as well as “unusual obsession with numbers“ and “visual hypersensitivity” were also observed. Twenty months after his initial visit, he began treatment with oral aripiprazole. His medication regimen was subsequently changed to venlafaxine hydrochloride, duloxetine hydrochloride, and risperidone, escitalopram oxalate and risperidone, and mirtazapine and risperidone, respectively. However, his symptom of pressure on his face did not change during the follow-up period. Because the patient expressed enthusiasm to undergo strabismus surgery, we performed medial rectus muscle recession (6.5 mm) of the right eye at 29 months after his first visit. Before the surgery, the alternate prism cover test revealed the angle of strabismus was 35 PD at near and 30 PD at distance. After the surgery, the angle of strabismus decreased at both near and distance. At 2 weeks after surgery, the angle of esotropia was 6 PD at near and 8 PD at distance. Thereafter, the angle of strabismus varied (12–30 PD at near and 6–18 PD at distance). Forty-one months after his first visit, the evaluation of microfluctuations in accommodation with Speedy-K (Righton, Tokyo, Japan) revealed a pattern consistent with SNR (Figure 1A). Refraction in both eyes based on the Spot Vision Screener (SVS) (Welch Allyn, Auburn, NY, USA) was +0.25 D in the right eye and +0.75 D in the left eye. Refraction with the fellow eye occluded based on the SVS was +0.75 D in the right eye and +1.0 D in the left eye. Pupil diameter in both eyes based on the SVS was 4.2 mm in the right eye and 4.3 mm in the left eye, and pupil diameter with the fellow eye occluded based on the SVS was 5.1 mm in the right eye and 4.1 mm in the left eye (Table 1). On the same day, the angle of strabismus was 18 PD at near and 12 PD at distance. Fifty-one months after his initial visit, botulinum toxin A injection (2.5 units) was administered to the left medial rectus muscle. Fifty-two months after his initial visit, the angle of strabismus was 6 PD at near and 4 PD at distance whereas Speedy-K revealed a pattern consistent with SNR (Figure 1B). The patient complained of motion sickness at distance. Fifty-five months after his initial visit, tropicamide and hydrochloride phenylephrine eye drops were started. Because Speedy-K findings remained unchanged at 57 months after his initial visit (Figure 1C) with persistent headache and dizziness but less motion sickness at distance, they were discontinued 60 months after his initial visit. Thereafter, he complained that his body felt pushed and distorted.

## 3. Discussion

Our patient presented with esotropia since childhood, but the angle of strabismus was variable, leading to a suspicion of SNR. We considered the condition of this patient to be SNR combined with esotropia since childhood. However, it is difficult to completely distinguish between concomitant esotropia and convergence spasm in this patient because the angle of strabismus has been reported to fluctuate in patients with infantile esotropia patients and poor stereopsis [6]. In addition to the convergence component, our patient had abnormal microfluctuations in accommodation, suggesting spasm involving the accommodation component. SNR is a condition in which one or more of the three components of the near reflex (convergence, accommodation, and pupil constriction) occurs intermittently [2]. Among 45 patients with SNR, two had only the convergence component [3]. The patient’s primary symptoms at the time of initial examination were headache with dizziness. We initially thought that these symptoms were due to changes in alternating visual images, but these symptoms, which are difficult to explain with strabismus alone, may lead us to suspect SNR. Patients with SNR present with intermittent episodes of diplopia, blurred vision, micropsia or macropsia, fluctuation of vision, headache, ocular pain, and dizziness [1]. Although we thought that wearing glasses with Fresnel membrane prisms could reduce the degree of change in alternating visual images, such use is unusual. We should be aware of the patient’s chief complaint.

There are four main causes of SNR: psychiatric disorders (e.g., stress, anxiety disorders, personality disorders), head trauma, organic diseases, and other [2]. Regarding the psychiatric causes of SNR in four patients with convergence spasm, two dated the onset of difficulty to the start of a new job that required prolonged close work, one attributed onset to the death of his father-in-law, and one thought that convergence spasm resulted from severe hyperemesis gravidarum during first pregnancy [7]. Our patient had ASD and somatic symptom disorder. We believe that our patient has had ASD since childhood, which was diagnosed after the visit to our hospital. Although the relationship of SNR to the pathology of ASD is unknown, it is possible that patients with ASD are more likely to develop SNR. In ASD, accommodative responses measured using the modified Nott dynamic retinoscopy technique are poor [4]. Although the pathophysiology of ASD remains undetermined, evidence of early life brainstem dysfunction in ASD have accumulated [8]. Some have speculated that neurological changes in the brainstem in ASD could result in reduced accommodative accuracy. The following near-point findings in children with ASD have been investigated in detail: distance and near phoria, near point convergence, near fusional convergence and divergence, accommodative response, and Northeastern State University College of Optometry Oculomotor Testing [5]. Patients with ASD have receded near point of convergence and poor fixation [5]. Accommodative response measured with monocular estimation method retinoscopy did not differ between individuals with ASD and typical development in one study [5].

Microfluctuations in accommodation are divided into a high-frequency component (1.0–2.3 Hz) and a low-frequency component (0.1–0.6 Hz) [9]. In accommodative spasm, high-frequency microfluctuations (1–4 Hz) in accommodation have been reported to occur more frequently [10]. Speedy series instruments are infrared optometers capable of analyzing microfluctuations in accommodation [11]. The Speedy-K optometer, which was used in our patient, revealed a pattern of accommodative spasm.

Treatment for SNR includes psychiatric evaluation and treatment, placebo, cycloplegic eye drops, plus lenses, miotic eye drops, and minus lenses for accommodative spasm. For convergence spasm, treatment involves botulinum toxin injection and prism glasses [2]. For our patient, strabismus surgery was performed before the diagnosis of SNR. After the diagnosis of SNR, botulinum toxin A injection and cycloplegic eye drops were administered. Unfortunately, both treatments were ineffective.

There are a few reports describing a patient with SNR that was difficult to differentiate from acute esotropia [12,13]. A 23-year-old male with acute adult onset concomitant esotropia and an 11-year-old boy with large-angle concomitant alternating esotropia were described. They had no limitations in eye movements, normal light reflexes, and myopia alleviated with cycloplegia. These patients were diagnosed as having SNR [12,13]. Our patient and these patients had similar clinical findings. In patients with SNR, the apparent limitations of abduction in version will disappear in duction [1]. Our patient had no limitations of abduction in either version or duction. Previous reports did not describe eye movements in duction or version [12,13]. The causes of acquired concomitant esotropia in adults might include strabismus associated with mechanical and age-related changes in the orbital pulley, such as strabismus associated with severe myopia and sagging eye syndrome, respectively [14,15]. In addition to these conditions, SNR is a rare cause of acquired concomitant esotropia. Because of the paucity of reports on SNR mimicking acute esotropia, the incidence of SNR with esotropia is unknown.

Since this study was a case report, it was difficult to reveal the relationship between SNR and ASD. In the future, we plan to investigate the incidence of SNR among patients with ASD and the pathogenesis of accommodative dysfunction in patients with ASD.

## 4. Conclusions

Our patient presented with concomitant esotropia, but the angle of strabismus was variable. We suspected SNR; evaluation of the patient’s accommodation confirmed the diagnosis. In addition, our patient’s main symptoms were hard to explain based on strabismus alone. In a patient with concomitant esotropia, if the symptoms are not exclusively due to strabismus, SNR should be suspected. Our patient had been diagnosed with ASD. Although the relationship between SNR and the pathology of ASD is unknown, it has been reported that patients with ASD have poor accommodative responses [4]. It is possible that patients with ASD are more likely to develop SNR.

## Figures and Tables

**Figure 1 reports-06-00038-f001:**
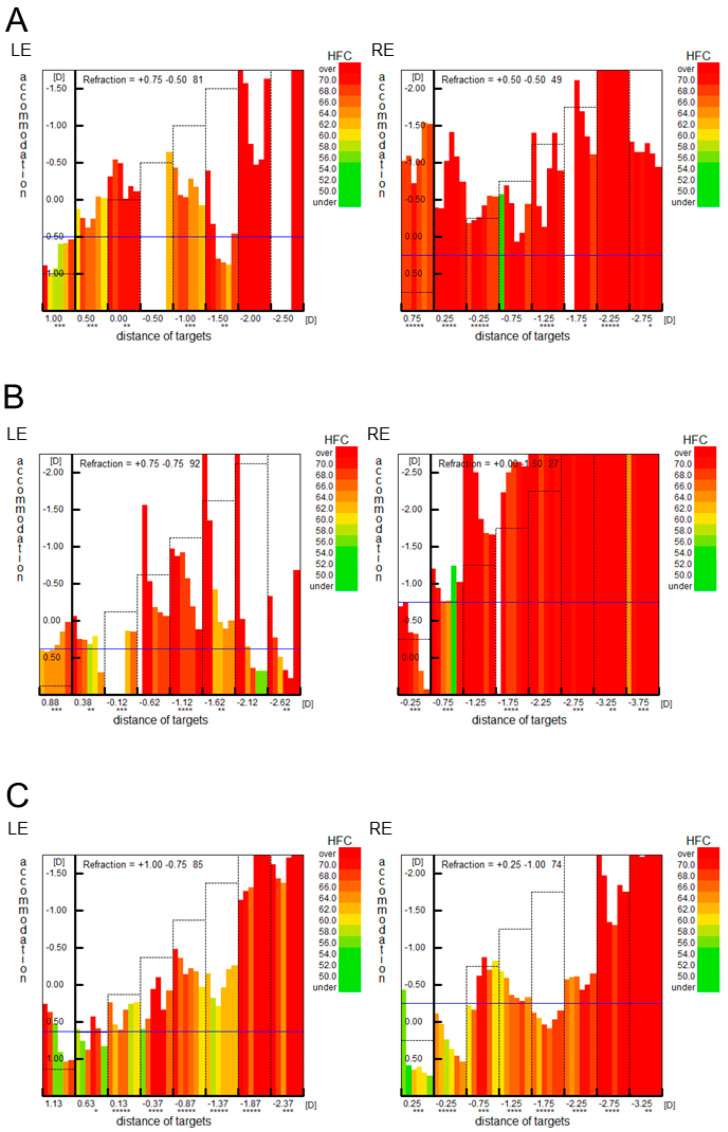
(**A**) Microfluctuations in accommodation of our patient analyzed with Speedy K (Righton, Tokyo, Japan) at 41 months after his initial visit. High-frequency components (HFC) are observed in both eyes, indicating a pattern of accommodative spasm. (**B**) Microfluctuations in accommodation at 52 months after his initial visit. (**C**) Microfluctuations in accommodation at 57 months after his initial visit. * represents reliability of measurements. Many asterisks represent high reliability. LE, left eye; RE, right eye; D, diopter.

**Table 1 reports-06-00038-t001:** Refraction and pupil diameter based on the Spot Vision Screener (Welch Allyn, Auburn, NY, USA) at 41 months after our patient’s initial visit.

	Right Eye	Left Eye
Refraction in both eyes	+0.25 D	+0.75 D
Refraction with fellow eye occluded	+0.75 D	+1.0 D
Pupil diameter in both eyes	4.2 mm	4.3 mm
Pupil diameter with fellow eye occluded	5.1 mm	4.1 mm

D, diopter.

## Data Availability

The data presented in this case report are available on request from the corresponding author.

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
