# Peer review of "Spasm of Near Reflex in a Patient with Autism Spectrum Disorder: A Case Report"

_reports, 2023, doi:10.3390/reports6030038_

Round 1

Reviewer 1 Report

The manuscript tells us a diagnostic process beginnimg with SNR to the final diagnosis of ASD. However the authors did not emphasize the important aspects sufficiently. 

1. The introduction should be modified. There is a study to indicate the ASD and accomodation problems. It should be cited and discussed. "https://www.ncbi.nlm.nih.gov/pmc/articles/PMC8051934/"

2. The possible known causes of SNR should be summarized in the introduction.

3. Discussion: The important aspect of the case, the process from SNR to ASD should be emphasized. 

b. The diagnostic procedures of ASD, and its symptoms should be discussed.

c. The above mentioned study should be cited and discussed. 

Author Response

July 27, 2023

Ms. Clair Li

Assistant Editor

Reports

Dear Ms. Li:

We wish to re-submit the manuscript titled "Spasm of near reflex in a patient with autism spectrum disorder: a case report". The manuscript ID is reports-2445003.

We appreciated the comments and suggestions provided by the reviewers. The manuscript has benefited from their insightful suggestions. I look forward to working with you and the reviewers to move this manuscript closer to publication in Reports.

The manuscript has been rechecked and the necessary changes have been made in accordance with the reviewers’ suggestions. The point-by-point responses to all comments are given below. The changes in the revised manuscript are shown in red font. The revised manuscript was checked by an editing service.

We hope that the revised manuscript is acceptable for publication in Reports. Thank you for your time and consideration. I look forward to hearing from you.

Sincerely,

Satoshi Ueki, MD, PhD

Division of Ophthalmology and Visual Science, Graduate School of Medical and Dental Sciences, Niigata University

Asahimachi-dori 1-757, Chuo-ku

Niigata 951-8510, Japan

Tel.: (+81)-25-227-2296

RESPONSES TO THE REVIEWER’S COMMENTS

Reviewer #1

Comment 1: The introduction should be modified. There is a study to indicate the ASD and accomodation problems. It should be cited and discussed. "https://www.ncbi.nlm.nih.gov/pmc/articles/PMC8051934/"

Response: We thank the reviewer for the constructive suggestions. We have cited the study. We have added the following sentences to the Introduction:

Although there are no reports describing a patient with both SNR and ASD and the incidence of SNR in patients with neurodevelopmental disorders such as ASD is undetermined, reports have described a relationship between near reflex and ASD [4-5]. We discuss the relationship between SNR and ASD.

We have also added the following sentences to the Discussion:

The following near-point findings in children with ASD have been investigated in detail: distance and near phoria, near point convergence, near fusional convergence and divergence, accommodative response, and Northeastern State University College of Optometry Oculomotor Testing [5]. Patients with ASD have receded near point of convergence and poor fixation [5]. Accommodative response measured with monocular estimation method retinoscopy did not differ between individuals with ASD and typical development in one study [5].

By adding the paper on the previous discussion about ASD and accommodation problems, we were able to have a more in-depth discussion. We thank the reviewer again.

Comment 2: The possible known causes of SNR should be summarized in the introduction.

Response: We thank the reviewer for the constructive suggestion. We have added the following sentences to the Introduction:

Organic diseases include multiple sclerosis, Chiari malformation, neurofibroma, pituitary adenoma, labyrinthitis, elevated intracranial pressure, vestibular neuritis, metabolic encephalopathy, stroke, epilepsy, dorsal midbrain syndrome, and cerebral aneurysm [2]. Other included post laser in situ keratomileusis, intermittent exotropia, excessive or prolonged near work, instrument myopia, hereditary cause, post myelography, and dissociation of binocularity [2].

Comment 3: Discussion: The important aspect of the case, the process from SNR to ASD should be emphasized. 

Response: We thank the reviewer for the constructive suggestion. We believe that the patient had ASD since childhood and ASD was diagnosed after the visit to our hospital. We have added the following sentence to the Discussion section:

We believe that our patient has had ASD since childhood, which was diagnosed after the visit to our hospital.

Comment 4: The diagnostic procedures of ASD, and its symptoms should be discussed.

Response: We thank the reviewer for the constructive suggestions We have added the following sentences to the Case Presentation:

The patient was diagnosed in accordance with the Diagnostic and Statistical Manual of Mental Disorders, Fifth Edition (DSM-5) criteria for ASD by an experienced child psychiatrist. DSM-5 diagnostic criteria for ASD include persistent deficits in social interaction across multiple contexts, social communication impairments, and restricted, repetitive patterns of behavior. Specifically, the patient had "difficulty with interpersonal interactions since early childhood and was unable to make friends even as a student. Even during medical examinations, he had difficulty making eye contact with others, which prevented him from deepening his emotional interactions”. Episodes indicative of social communication impairements as well as "unusual obsession with numbers " and "visual hypersensitivity" were also observed.

Reviewer #2

Comment 1: In lines 37 and 39, the authors quote inappropriately. They say:…According to Hyndman, there are four main causes of SNR: psychiatric disorders (stress, anxiety disorders,…

The same happens when citing: …Roy et al. reported that among… y cuando citan a:…Papageorgiou et al. have described patients… Please review the entire manuscript, there are other authors misquoted.

Hyndman is not correctly cited. Follow the citation rules of the journal and renumber the authors.

Response: We thank the reviewer for the constructive suggestions We have carefully re-checked the entire manuscript and the citation rules of the journal. We have deleted “according to …” and “… et al. … ”. If our interpretation of the reviewer’s comment is wrong, please point it out.

Comment 2: The introduction of the manuscript is insufficient, they should talk about the incidence of the disorder in normal patients and in patients with neurodevelopmental disorders. Please argue and adequately justify the study in your introduction. Does the NRS have any relation with the incidence in patients with neurodevelopmental disorders?

Response: We thank the reviewer for the constructive suggestion. Unfortunately, there are no reports about the incidence of SNR in patients with neurodevelopmental disorders. We have added the following sentences about the incidence of SNR in normal patients to the Introduction:

The incidence of SNR in patients who visited a single binocular vision and orthoptics clinic was 5.1% [3]. …Although there are no reports describing a patient with both SNR and ASD and the incidence of SNR in patients with neurodevelopmental disorders such as ASD in undetermined, reports have described a relationship between near reflex and ASD [4,5]. We discuss the relationship between SNR and ASD.

Comment 3: Why did they choose this case and not another?

Response: We chose this case because it was rare due to the findings implying a relationship between SNR and ASD. The findings mimicked acquired esotropia. We have encountered other patients with SNR and ASD. However, we had not been able to perform long-term follow-up of those patients.

Comment 4: The authors should provide data on the incidence of patients presenting with strabismus and SNR.

Response: We thank the reviewer for the constructive suggestion. Because there are few reports about SNR mimicking strabismus, we cannot provide data on the incidence of patients with strabismus and SNR. We have added the following sentence to the Discussion:

Because of the paucity of reports describing SNR mimicking acute esotropia, the incidence of SNR with esotropia is unknown.

Comment 5: The authors should clarify the ethical aspects of the study. How was confidential information handled? Who authorized the study? Was it supervised and authorized by an ethics committee?

Response: We thank the reviewer for the critical comment. The patient himself authorized the study. This report does not contain any personal information that could lead to the identification of the patient. We have revised the Institutional Review Board Statement as follows.

This study was conducted in accordance with the Declaration of Helsinki. The Institutional Review Board at Niigata University determined that it met criteria for waiver due to these reasons: written informed consent was obtained from the patient, and this report did not include personally identifiable information.

Comment 6: When dealing with a patient with ASD, who authorized the study? The patient him/herself? Family members?

Response: The patient himself authorized the study. We have revised the Informed Consent Statement as follows. Written consent to publish this case report was obtained from the patient himself.

Comment 7: Authors should add a section clearly stating the objective and a section on limitations.

Response: We thank the reviewer for the constructive suggestion. We have added the following sentence clearly stating the objective of this study to the Introduction:

The objectives of this case report are to propose tips to accurately diagnose SNR and to present an in-depth discussion on whether patients with ASD might develop SNR.

Furthermore, we have added a paragraph about limitations of this study in the Discussion section:

Since this study was a case report, it was difficult to reveal the relationship between SNR and ASD. In the future, we plan to investigate the incidence of SNR among patients with ASD and the pathogenesis of accommodative dysfunction in patients with ASD.

To clearly describe our objectives, we have added the following sentence to the Abstract:

Although the relationship between SNR and the pathology of ASD is unknown, it is possible that patients with ASD are more likely to develop SNR.

We have also added the following sentences to the Conclusions:

Our patient had been diagnosed with ASD. Although between SNR and the pathology of ASD is unknown, it has been reported that patients with ASD have poor accommodative responses [4]. It is possible that patients with ASD are more likely to develop SNR.

The revised manuscript consists of 2,359 words.

Reviewer 2 Report

In lines 37 and 39, the authors quote inappropriately. They say:…According to Hyndman, there are four main causes of SNR: psychiatric disorders (stress, anxiety disorders,…

The same happens when citing: …Roy et al. reported that among… y cuando citan a:Papageorgiou et al. have described patients… Please review the entire manuscript, there are other authors misquoted.

Hyndman is not correctly cited. Follow the citation rules of the journal and renumber the authors.

The introduction of the manuscript is insufficient, they should talk about the incidence of the disorder in normal patients and in patients with neurodevelopmental disorders. Please argue and adequately justify the study in your introduction. Does the NRS have any relation with the incidence in patients with neurodevelopmental disorders?

Why did they choose this case and not another?

All these questions should be addressed by the authors.

The authors should provide data on the incidence of patients presenting with strabismus and SNR.

The authors should clarify the ethical aspects of the study. How was confidential information handled? Who authorized the study? Was it supervised and authorized by an ethics committee?

When dealing with a patient with ASD, who authorized the study? The patient him/herself? Family members?

Authors should add a section clearly stating the objective and a section on limitations.

We encourage the authors to make the pertinent changes.

Best regards

English should be reviewed by a native speaker.

Author Response

(The authors gave the same response as above.)

Round 2

Reviewer 1 Report

Thank you for the revisions.

Reviewer 2 Report

I thank the authors for the changes made. The manuscript can now be accepted. The changes made have greatly improved the quality of the manuscript.

Best regards

Minor changes to the manuscript are required